# Suppression of DNCB-Induced Atopic Skin Lesions in Mice by *Wikstroemia indica* Extract

**DOI:** 10.3390/nu12010173

**Published:** 2020-01-08

**Authors:** So-Yeon Lee, No-June Park, Jonghwan Jegal, Beom-Geun Jo, Sangho Choi, Sang Woo Lee, Md. Salah Uddin, Su-Nam Kim, Min Hye Yang

**Affiliations:** 1College of Pharmacy, Pusan National University, Busan 46241, Korea; s7315@naver.com (S.-Y.L.); jhjegal@pusan.ac.kr (J.J.); dtc98103@pusan.ac.kr (B.-G.J.); 2Natural Products Research Institute, Korea Institute of Science and Technology, Gangneung 25451, Korea; 115519@kist.re.kr; 3International Biological Material Research Center, Korea Research Institute of Bioscience and Biotechnology, Daejeon 34141, Korea; decoy0@kribb.re.kr (S.C.); ethnolee@kribb.re.kr (S.W.L.); 4Ethnobotanical Database of Bangladesh, Dhaka 1208, Bangladesh; Plantsofbd@gmail.com

**Keywords:** *Wikstroemia indica*, 2,4-dinitrochlorobenzene, atopic dermatitis, transepidermal water loss, interleukin

## Abstract

*Wikstroemia indica* (L.) C.A. Mey. is used in traditional Chinese medicine to treat inflammatory diseases such as arthritis and bronchitis. In this study, we aimed to investigate the effects of an ethanolic extract of *W. indica* on cutaneous inflammation in mice with 2,4-dinitrochlorobenzene (DNCB)-induced atopic dermatitis (AD). Dermal administration of *W. indica* ethanolic extract to DNCB-sensitized hairless mice with dermatitis, for two weeks, reduced erythema, scaling, and edema. Skin hydration was improved and transepidermal water loss was reduced at a *W. indica* concentration of 1%. Furthermore, *W. indica* also significantly reduced serum IgE and IL-4 concentrations in our mouse model. These results suggest that *W. indica* has potential as a topical treatment for AD and as an adjunctive agent to control AD.

## 1. Introduction

Atopic dermatitis (AD) is a chronic, relapsing inflammatory disease that is accompanied by representative symptoms, such as itching, redness, dry skin, and exudate. According to the Korean National Health and Nutrition Examination Survey (2016), the prevalence of AD in Korean adolescents was 25.1% [1]. In reality, children and adolescents account for more than half of AD patients, though in recent years, the proportion of adult AD patients has increased [2,3]. AD is a multifactor disorder caused by complex interactions between immunological, environmental, and genetic factors [4,5], and its hallmarks are markedly elevated serum immunoglobulin E (IgE) and interleukin-4 (IL-4) levels, and a damaged skin barrier function [6,7].

The skin is the first-line of defense system against external allergens, pollutants, and microbes, and maintains water homeostasis in the stratum corneum [8]. Skin damage increases the susceptibility to penetration of the skin barrier by foreign substances or microorganisms, and because many of these exogenous stressors are potential allergens, skin penetration can result in allergic reactions. Furthermore, allergic reactions increase the secretions of cytokines, which exacerbate skin barrier damage and initiate a vicious cycle [9]. Traditionally, AD is regarded as an allergic disease with an immunological etiology, and accordingly, immunomodulators such as topical corticosteroids and calcineurin inhibitors have been mainly used as therapeutic agents [10,11]. However, recent studies have shown that skin barrier damage per se might be a major cause of AD, and this has resulted in novel pharmacotherapeutic developmental strategies [12]. For example, emollients provide a single barrier layer on skin and provide a safe and effective means of reducing skin penetration by allergens and maintaining skin hydration. Some clinical trials have shown emollients help AD in newborns without causing any side effects [13,14].

*Wikstroemia indica* (L.) C.A. Mey. (Thymelaeaceae) is a shrub species of flowering plants that are native to Southeast Asia. *W. indica* has long been used in traditional Chinese medicine for treating arthritis, bronchitis, and syphilis [15], and scientific studies have shown that *W. indica* contains bioactive compounds (e.g., lignans, flavonoids, and coumarins) [16,17]. In addition, extracts or isolated compounds have been shown to have antitumor [18], anti-malarial [19], anti-inflammatory [20], antifungal, antimitotic, and antiviral [21] effects. However, no study has investigated the anti-AD effects of *W. indica*, and thus, in the present study, we investigated the anti-AD activity of *W. indica* ethanolic extract in a 2,4-dinitrochlorobenzene (DNCB)-induced murine model of atopic dermatitis.

## 2. Materials and Methods

### 2.1. Plant Materials and Sample Preparation

The stem and leaf parts of *Wikstroemia indica* (L.) C.A. Mey were collected in Zobra village, Hathazari Upazila, Chittagong, and the plant materials were authenticated by Dr. Sang Woo Lee (Korea Research Institute of Bioscience and Biotechnology) in 2014. A voucher specimen (accession number KRIB 0055091) of the retained material is preserved at the herbarium of KRIBB. The plant name and family name of *Wikstroemia indica* (L.) C.A. Mey. was checked with The Plant List on-line site (www.theplantlist.org; http://www.theplantlist.org/tpl1.1/record/tro-32000565) and was added to the Plant List (www.theplantlist.org). A voucher specimen (PNU-0026) was deposited at the Medicinal Herb Garden of Pusan National University. Dried stems and leaves of *W. indica* (4 kg) were prepared by chopping them into small millimeter-sized pieces. The plants prepared were extracted with 95% EtOH (12 L × 3) and concentrated to dryness in vacuo to yield *W. indica* EtOH extract (247.6 g).

### 2.2. Mouse Model

SKH-1 hairless mice (female, six weeks old) were acquired from Orient Bio Inc. (Seongnam, Korea). Mice were housed in an air-controlled environment (RH 55 ± 5%, 25 ± 5 °C) under 12 h light and 12 h dark cycle until the experiment, and were provided with standard laboratory food and water ad libitum. All experimental procedures involving animals complied with the Guide for the Care and Use of Laboratory Animals of the National Institutes of Health (NIH Publication No. 85-23, 2011 revision). Procedures were approved by the Institutional Animal Care and Use Committee (IACUC) of Korea Institute of Science and Technology (Certification No. KIST-2016-011).

### 2.3. DNCB-Induced AD Model and W. indica Extract Treatment

Atopic dermatitis (AD) was induced in SKH-1 hairless mice using DNCB (2,4-dinitrochlorobenzene; Sigma-Aldrich, Seoul, Korea), based on previous literatures [22,23]. Briefly, the mice were acclimatized for one week and then divided into four groups (n = 7/group), as follows; distilled water-treated controls (the CON group), DNCB-sensitized and vehicle-treated controls (DNCB + vehicle group), a DNCB-sensitized and a 1% *W. indica* extract treatment group (the DNCB + *W. indica* group), and a DNCB-sensitized and a 1% pimecrolimus (Elidel^®^) treated group (the DNCB + Elidel group). To induce AD-like skin lesions, 1% DNCB (100 μL) dissolved in propylene glycol and EtOH (7:3) was topically applied to dorsal skin, daily, for 8 days (days 0 to 7). Mice were challenged with 0.1% DNCB (100 μL) at 2-day intervals for 2 weeks (on days 8, 10, 12, 14, 16, 18, and 20). Animals in the DNCB + *W. indica* group and in the DNCB + Elidel group were treated topically with 1% *W. indica* (100 μL) or 1% pimecrolimus cream twice daily, over the same two-week period, 4 h before DNCB administration. The eczema reaction and the degree of erythema, exudation, and excoriation were evaluated clinically. Mice were sacrificed and the dorsal skin tissue was excised after the final experiment (day 21) for histological examination. Blood was collected from abdominal aortas to determine serum IgE and IL-4 levels.

### 2.4. Histological Analysis

Dorsal skin tissues were collected from the SKH-1 hairless mice for histological analysis on day 21. Tissues were fixed in 10% formalin solution for 24 h and embedded in paraffin wax. Sections (2–3 mm thick) were obtained using a microtome, placed on slides, and dried overnight at 37 °C. Staining was performed using hematoxylin and eosin (H & E) to determine changes in thickness of the epidermis or toluidine blue, which would indicate the changes in mast cell number. The slides were observed and photographed under a light microscope (Olympus CX31/BX51, Olympus Optical Co., Tokyo, Japan) and a fluorescence microscope (TE2000-U, Nikon Instruments Inc., Melville, NY, USA).

### 2.5. Measurement of IgE and IL-4 Levels by ELISA

Total levels of serum IgE and IL-4 were measured using enzyme-linked immunosorbent assay kits (ELISA, eBioscience, San Diego, CA, USA). Blood samples were collected from abdominal aortas and centrifuged at 10,000 rpm for 15 min at 4 °C. The serum samples obtained were stored at −80 °C until required.

### 2.6. Measurement of Transepidermal Water Loss and Skin Hydration

The transepidermal water loss (TEWL) was measured using an evaporimeter (Tewameter TM210; Courage and Khazaka, Cologne, Germany). Skin hydration and skin pH were evaluated using a SKIN-O-MAT unit (Cosmomed, Ruhr, Germany). All measurements were performed weekly in an environment of 50 ± 5% relative humidity at 25 ± 5 °C temperature.

### 2.7. Analysis of HPLC/MS

High-performance liquid chromatography/mass spectrometry (HPLC/MS) analyses were carried out in Agilent 6530 Accurate-Mass Q-TOF LC/MS system (Agilent Technologies). The chromatographic separation was performed using a Poroshell 120 EC-C18 column (3.0 × 100 mm, 2.7 μm, Agilent Technologies, Little Fall, DE, USA), and the gradient elution used mobile phase A (acetonitrile) and B (ultrapure water). The gradient elution program was set as: 10% A (0–5 min), 10%–70% A (5–30 min), and 70% A (30–40 min). The flow rate was maintained at 0.3 mL/min, and UV detection at 254 nm. HPLC/MS spectra was acquired in positive ionization mode with a mass range of 100–1,500 m/z.

### 2.8. Statistical Analysis

Results were obtained using at least two independent experiments. The data were analyzed by one-way analysis of variance (ANOVA) and Tukey’s multiple comparisons post-hoc analysis. Results were expressed as means ± SEMs and were considered to be statistically significant for *p*-values < 0.05.

## 3. Results

### 3.1. Effects of W. indica Extract on AD-Like Symptoms in the DNCB Hairless Mouse Model

To investigate anti-atopic effects of *W. indica* on DNCB-sensitized skin lesions, dermatitis severities were evaluated using skin lesion images. The procedure used to establish the DNCB-induced AD murine model is shown in Figure 1A. Severe AD-like symptoms, i.e., dried skin, cornification, exudation, and erythema, were observed on the dorsal skins of hairless mice after 3 weeks exposure to DNCB. The application of 1% *W. indica* during the 2-week challenge period (days 8–21) significantly relieved AD-like symptoms (Figure 1B).

### 3.2. Effects of W. indica on Epidermal Thickness in DNCB-Induced Atopic Mice

To determine the histopathological features, the epidermal tissue of the dorsal skin was stained with H & E. Staining sections revealed that the epidermal tissue of the dorsal skin was about 2.5 times thicker in the DNCB (75.25 μm) than in the CON (Figure 2A), dorsal skin dermal thicknesses were 33.5% and 44.4% lower in the DNCB + *W. indica* and DNCB + Elidel groups than in the DNCB group (Figure 2B).

### 3.3. Effects of W. indica on Mast Cell Infiltration in DNCB-Induced Atopic Mice

Toluidine blue staining was applied to AD-affected dorsal skin tissue samples to observe changes in mast cell numbers. The toluidine blue mast stained cell numbers were greater in the DNCB than in the CON (Figure 3A), but were 38.3% and 33.3% lower in the DNCB + *W. indica* and DNCB + Elidel, respectively, than in the DNCB (Figure 3B).

### 3.4. Effects of W. indica Extract on Serum IgE and IL-4 Levels in DNCB-Induced Atopic Mice

Serum level of IgE was 4.9-fold higher and IL-4 was 3.76-fold higher in the DNCB than in the CON but 33% and 34.2% lower, respectively, in the DNCB + *W. indica* than in the DNCB (Figure 4A,B, respectively).

### 3.5. Effects of W. indica Extract on Skin Barrier Function in DNCB-Induced Atopic Mice

TEWL and skin hydration, which are quantitative indicators of skin barrier function, were both found to deteriorate rapidly in the DNCB group. More specifically, TEWL was approximately 4-fold higher in the DNCB than in the CON (103.15 vs. 25.08 g/m^2^/h) (Figure 5A). In addition, skin hydration was 70% lower in the DNCB (13.68% vs. 47.58, respectively) (Figure 5B). However, TEWL was reduced to 21.7% in the DNCB + *W. indica* group (Figure 4A) and skin hydration was improved to 17.83% in the DNCB + *W. indica* group (Figure 4B).

### 3.6. The Phytochemical Characterization of W. indica Using the High-Performance Liquid Chromatography/Mass Spectrometry (HPLC/MS)

A HPLC/MS method for analyzing phenolic compounds in *W. indica* EtOH extract was developed. For the simultaneous determination of the main compounds of *W. indica*, the optimum analytical condition was investigated. The optimal mobile phase, which consisted of acetonitrile/water, was subsequently employed for the analysis of *W. indica* and led to a good resolution and satisfactory peak shape. The presence of three compounds, 1: umbelliferone (m/z 161.0264 at *t*_R_ 12.368 min), 2: quercitrin (m/z 447.0994 at *t*_R_ 22.718 min), and 3: daphnoretin (m/z 351.0562 at *t*_R_ 27.377 min) in *W. indica* was confirmed by a comparison of its UV spectrum and retention time of each standard compound (Figure 6A,B).

## 4. Discussion

*Wikstroemia* species are still used as a herbal traditional medicines in China to treat various inflammatory conditions [17,24]. *W. indica*, a member of the *Wikstroemia* genus, is listed in Chinese pharmacopoeia under the name ‘liao ga wang’ and is marketed as an over-the-counter product, as an anti-inflammatory agent [24]. In recent years, many studies have demonstrated the anti-inflammatory and anti-viral effects of the members of the *Wikstroemia* genus [17,25]. Furthermore, *W.*
*indica* has been shown to contain many phytochemicals, such as flavonoids, lignans, and coumarins, which have all got potent anti-inflammatory effects [25,26].

Previous studies on plants of the genus *Wikstroemia* have shown that its crude extracts have immunomodulatory, anti-inflammatory, and anti-allergic effects in murine models [17,20,21]. However, no systematic attempt has been made to determine the effect of *W. indica* on AD, and thus, we investigated the anti-atopic effects of the airborne parts of *W. indica* ethanolic extract in a DNCB-induced model of AD in hairless mice. We found 1% *W. indica* markedly attenuated DNCB-sensitized scratching behavior and skin lesion severity, such that it prevented the increase of TEWL and reduction of skin hydration induced by DNCB. A reduced epidermal barrier function is considered to be a major predisposing factor of AD, and thus, the use of moisturizers offers a first-line therapy for the management of mild AD [27,28]. From this point of view, it is possible that *W. indica* extract helps maintain the skin barrier function, and thus, ameliorated atopic skin symptoms. Future studies are required to evaluate the moisturizing effects of *W. indica* extract in patients with AD.

According to our findings, *W. indica* extract markedly inhibited IL-4 overexpression in the serum of our murine model. The inhibitory effects of *W. indica* on atopic skin symptoms might be in part due to the decrease in IL-4 and IgE levels. IL-4 inhibition might prevent the differential development of Th0 into Th2 cells, IgE class switching, and the degranulation of mast cells by IgE, which helps lower allergic responses [29]. Serum IgE level elevations in AD patients are influenced by activation and the secretion of IL-4 by Th2 cells (type 2 helper T cells) [30], and it has been reported that IL-4 acts as an essential factor that promotes B cells to the IgE pathway [31,32]. Recent studies have shown that IL-4 enhances the productions of chemokines, such as MCP-1 and eotaxin, which increase the infiltrations of Th2 cells, eosinophils, and macrophages into skin lesion [33,34,35]. IL-4 also contributes to the impairment of barrier homoeostasis, and the proteins filaggrin, loricrin, and involucrin produced by keratinocytes play an essential role in skin barrier recovery. The genes of these proteins are down-regulated when IL-4 is overexpressed and this prevents the skin barrier from functioning properly [36,37]. Therefore, we suggest treatment with *W. indica* extract might accelerate skin barrier recovery in AD-like skin lesions by suppressing IL-4 secretion by Th2-type cells.

Collectively, the present study showed that the ethanolic extract of *W. indica* significantly suppressed DNCB-sensitized AD, by reducing epidermal thickness and mast cell infiltration in our murine model. *Wikstroemia indica* extract showed decreases in serum IgE and the suppression of IL-4 levels in DNCB-induced AD mice. In addition, *W. indica* extract markedly suppressed DNCB-induced skin barrier impairment and skin dehydration in our AD murine model. Furthermore, this study provided evidence in support of the protective effects of *W. indica* for the treatment of inflammatory skin diseases.

## Figures and Tables

**Figure 1 nutrients-12-00173-f001:**
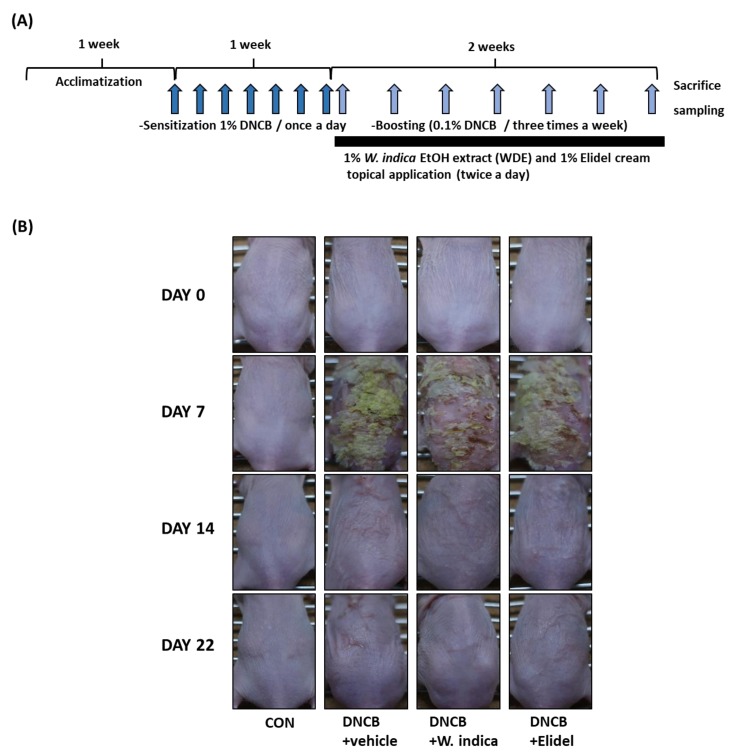
Effects of *W. indica* on the development of 2,4-dinitrochlorobenzene (DNCB)-induced AD-like skin lesions in SKH-1 hairless mice. (**A**) Schematic representation of the experiment. (**B**) Clinical features of DNCB-induced AD-like skin symptoms. CON: vehicle control group; DNCB + vehicle: DNCB-treated control group; DNCB + *W. indica*: DNCB plus 1% *W. indica*-extract treated group; and DNCB + Elidel: DNCB plus 1% pimecrolimus treated group.

**Figure 2 nutrients-12-00173-f002:**
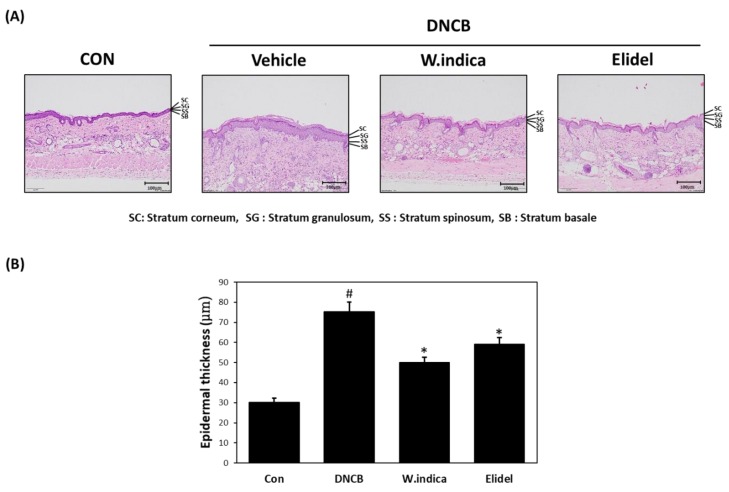
Effects of *W. indica* extract on hematoxylin and eosin (H & E) histopathology findings and epidermal thickness. (**A**) H & E staining result. (**B**) Epidermal thickness. Results are expressed as the means ± SEMs of two independent experiments. ^#^
*p* < 0.05 vs. the CON group; * *p* < 0.05 vs. the DNCB group. CON: vehicle control group; DNCB: DNCB + vehicle treated group; DNCB + *W. indica*: DNCB and 1% *W. indica* extract treated group; and DNCB + Elidel: DNCB and 1% pimecrolimus treated group.

**Figure 3 nutrients-12-00173-f003:**
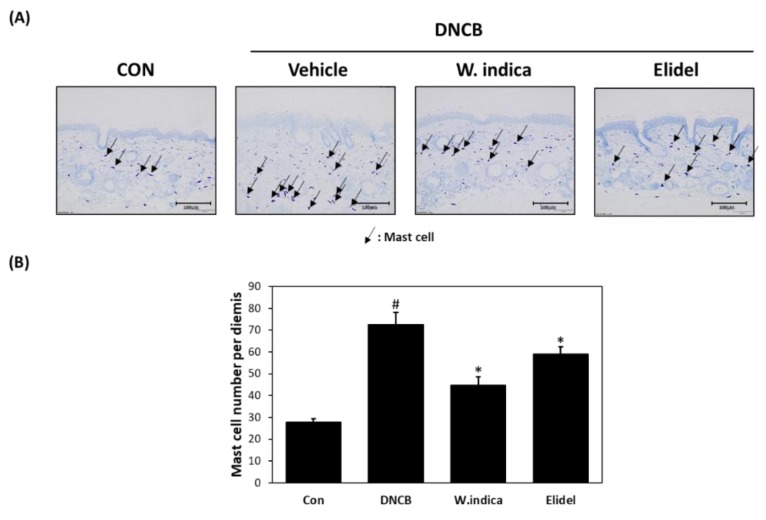
Effects of *W. indica* extract on toluidine blue staining findings and mast cell numbers in dermal tissues. (**A**) Toluidine blue staining result. (**B**) Mast cell numbers. Results are expressed as the means ± SEMs of two independent experiments. ^#^
*p* < 0.05 vs. the CON group; * *p* < 0.05 vs. the DNCB group. CON: vehicle control group; DNCB: DNCB + vehicle treated group; DNCB + *W. indica*: DNCB and 1% *W. indica* extract treated group; and DNCB + Elidel: DNCB and 1% pimecrolimus treated group.

**Figure 4 nutrients-12-00173-f004:**
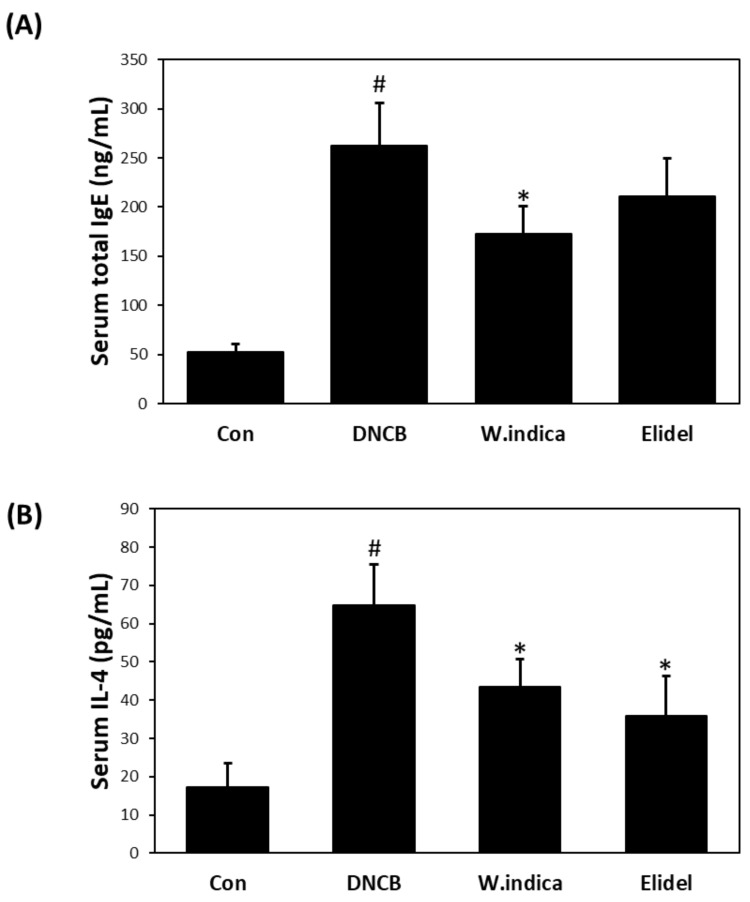
Effects of *W. indica* on serum IgE and IL-4 levels. (**A**) Serum IgE levels. (**B**) Serum IL-4 levels. Results are expressed as the means ± SEMs (n = 7) of two independent experiments. ^#^
*p* < 0.05 vs. the CON group; * *p* < 0.05 vs. the DNCB group. CON: vehicle control group; DNCB: DNCB + vehicle treated group; DNCB + *W. indica*: DNCB and 1% *W. indica* extract treated group; and DNCB + Elidel: DNCB and 1% pimecrolimus treated group.

**Figure 5 nutrients-12-00173-f005:**
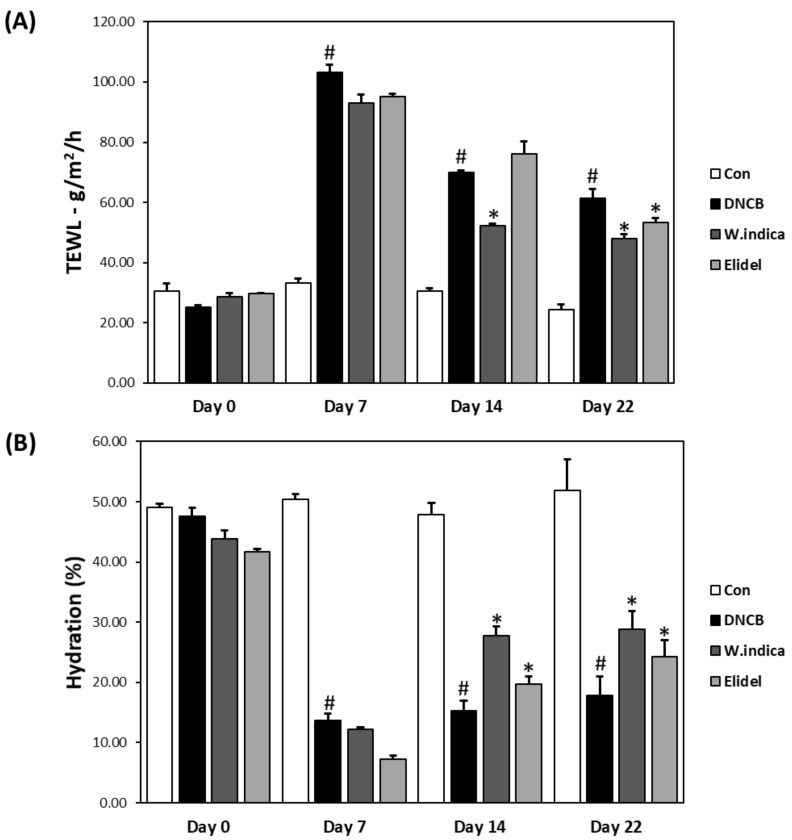
Effects of *W. indica* extract on skin barrier function. (**A**) Transepidermal water loss (TEWL). (**B**) Skin hydration values. Results are expressed as the means ± SEMs (n = 7) of two independent experiments. ^#^
*p* < 0.05 vs. the CON group; * *p* < 0.05 vs. the DNCB group. CON: vehicle control group; DNCB: DNCB + vehicle treated group; DNCB + *W. indica*: DNCB and 1% *W. indica* extract treated group; and DNCB + Elidel: DNCB and 1% pimecrolimus treated group.

**Figure 6 nutrients-12-00173-f006:**
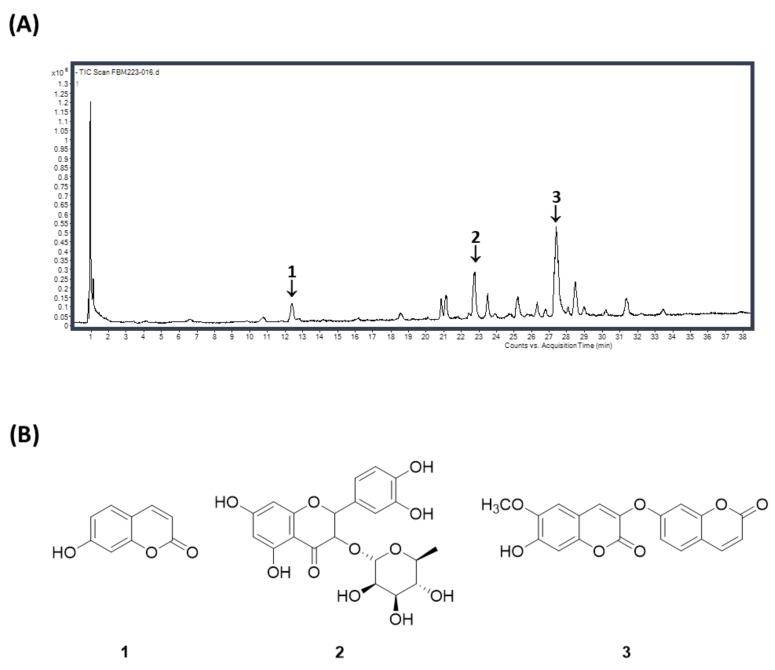
HPLC chromatogram of the phenolic compounds of *Wikstroemia indica* (**A**) and the chemical structures of its major compounds (**B**). Phytochemical characterization of the *W. indica* EtOH extract was performed using HPLC/MS. **1**: umbelliferone (*t*_R_ 12.368 min), **2**: quercitrin (*t*_R_ 22.718 min), and **3**: daphnoretin (*t*_R_ 27.377 min).

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
