# Peer review of "Suppression of DNCB-Induced Atopic Skin Lesions in Mice by Wikstroemia indica Extract"

_nutrients, 2020, doi:10.3390/nu12010173_

Round 1

Reviewer 1 Report

This manuscript have a lot of problems to be addressed.

1.Abstract

The end of the abstract, the authors refer to allergic contact dermatitis, which is incorrect, may be atopic dermatitis.

TEWL and skin hydration were also suppressed.....

This is incorrect. Correctly,
TEWL was reduced and skin hydration was improved...

2. Methods

The authors said that their model is atopic dermatitis model. If so, the authors should cite certain papers showing the evidence that the model is established, internationally accepted atopic dermatitis model.

The authors should show which points are evaluated histologically and clinically in methods section not just in results section.

like mast cell number, epidermal thickness in histology, or erythema, exudation, dried skin, cornification clinically.

Moreover, the authors should show the clinical scores mimicking EASI, not just showing photos.

The authors should evaluate CD3+T cell number to see if wilkstroemia extract reduce the infiltration, and also evaluate serum IL-13 levels.

3.Results3.5

The description for TEWL or skin hydration is incorrect.

Correctly,

TEWL was reduced to 21.7% in DNCB+W and skin hydration was improved to 28.8% in DNCB +W compared to 17.83% in DNCB group.

4. Discussion

2nd paragraph

The description should be corrected like

’the increase of TEWL and reduction of skin hydration induced by DNCB’

3rd paragraph 

Please describe the names of chemokines induced by IL-4.

The usage of the word ’however’ is not in accordance with the context, and should be deleted, or revise this sentence.

The authors should describe concretely which components like flavonoid, lignin, coumarine, may improve the skin eruption or reduce IL-4 expression citing relevant papers.

The authors should describe the possible mechanisms how witkstromia extract reduce IL-4 expression, are there previous studies showing such effects? If so, please cite such papers.

4th paragraph

The authors said that the reduction of skin thickness or mast cell number is dependent on decrease of IgE and IL-4.

This is overestimation. There is no evidence showing such dependency. It is  possible that these phenomena may just occur in parallel. Thus please revise this expression.

Author Response

Abstract

The end of the abstract, the authors refer to allergic contact dermatitis, which is incorrect, may be atopic dermatitis. TEWL and skin hydration were also suppressed..... This is incorrect. Correctly, TEWL was reduced and skin hydration was improved...

- We appreciate reviewer’s valuable comment and now we revised it as suggestion (Please see yellow-highlighted parts).

Methods

The authors said that their model is atopic dermatitis model. If so, the authors should cite certain papers showing the evidence that the model is established, internationally accepted atopic dermatitis model.

- We added two proper papers showing the evidence that the DNCB model is established for animal experiment of atopic dermatitis (Please see yellow-highlighted parts).

The authors should show which points are evaluated histologically and clinically in methods section not just in results section. like mast cell number, epidermal thickness in histology, or erythema, exudation, dried skin, cornification clinically.

- We newly added ‘The eczema reaction and the degree of erythema, exudation, and excoriation were evaluated clinically.’ In 2.3. DNCB-induced AD model and W. indica extract treatment and changed ‘Staining was performed using hematoxylin and eosin (H&E) or toluidine blue’ to ‘Staining was performed using hematoxylin and eosin (H&E) to determine changes in thickness of the epidermis or toluidine blue to determine changes in mast cell number.’ in 2.4. Histological analysis as suggested by reviewer.

Moreover, the authors should show the clinical scores mimicking EASI, not just showing photos.

- Unfortunately, we only checked changes in clinical skin symptoms by taking a picture and measuring epidermal thickness and mast cell numbers using dermal tissues at this time. The method of clinical scores mimicking EASI has not been performed.

The authors should evaluate CD3+T cell number to see if wilkstroemia extract reduce the infiltration, and also evaluate serum IL-13 levels.

- We appreciate reviewer’s valuable comment. Due to lack of serum sample, we only checked IL-4 and IgE serum concentrations at this time.

Results3.5

The description for TEWL or skin hydration is incorrect. Correctly, TEWL was reduced to 21.7% in DNCB+W and skin hydration was improved to 28.8% in DNCB +W compared to 17.83% in DNCB group.

- We changed it to ‘However, TEWL was reduced to 21.7% in the DNCB + W. indica group (Fig. 4A) and skin hydration was improved to 17.83% in the DNCB + W. indica group (Fig. 4B).’ as suggested by reviewer.

Discussion

2nd paragraph

The description should be corrected likethe increase of TEWL and reduction of skin hydration induced by DNCB’

- It is now changed to ‘We found 1% W. indica significantly alleviated DNCB-induced skin lesion severity, dermatitis scores, and scratching behavior, and that it prevented the increase of TEWL and reduction of skin hydration induced by DNCB.’.

3rd paragraph

Please describe the names of chemokines induced by IL-4.

- It is now added such as ‘Recent studies have shown IL-4 enhances the productions of chemokines, such as MCP-1 and eotaxin, that increase the infiltrations of inflammatory cells such as Th2 cells, eosinophils, and macrophages into lesion skin’ with more proper Ref. 36.

The usage of the word ’however’ is not in accordance with the context, and should be deleted, or revise this sentence.

- We deleted the word ‘however’ as suggested by reviewer.

The authors should describe concretely which components like flavonoid, lignin, coumarine, may improve the skin eruption or reduce IL-4 expression citing relevant papers.

- In our revised version of manuscript, we performed the HPLC/MS study for phytochemical analysis of W. indica extract and the result is newly added (Please see 3.6. The phytochemical characterization of W. indica using the high-performance liquid chromatography/mass spectrometry (HPLC/MS) in Results and Figure 6.). According to the result of chemical profiling, two coumarins (umbelliferone and daphnoretin) and a flavonoid (quercitrin) are major components of W. indica, and those compounds might improve the skin eruption or reduce IL-4 expression.

The authors should describe the possible mechanisms how witkstromia extract reduce IL-4 expression, are there previous studies showing such effects? If so, please cite such papers.

- We described a possible mechanism how W. indica extract reduces IL-4 expression such as ‘According to our findings, W. indica extract markedly inhibited IL-4 overexpression in serum of our murine model. The inhibitory effects of W. indica on atopic skin symptoms may be due in part to the decrease in IL-4 and IgE levels. IL-4 inhibition may prevent the differential development of Th0 into Th2 cells, IgE class switching, and the degranulation of mast cells by IgE, which mediates lowering allergic response’ and cited our previous literature as Ref. 29.

4th paragraph

The authors said that the reduction of skin thickness or mast cell number is dependent on decrease of IgE and IL-4. This is overestimation. There is no evidence showing such dependency. It is possible that these phenomena may just occur in parallel. Thus please revise this expression.

- We changed it to ‘Collectively, the present study shows that ethanolic extract of W. indica significantly suppressed DNCB-sensitized AD by reducing epidermal thickness and mast cell infiltration in our murine model. Wikstroemia indica extract showed decreases in serum IgE and the suppression of IL-4 levels in DNCB-induced AD mice.’.

Reviewer 2 Report

1. Why did the authors use only Female mice? 

2. The author should discuss the active components and possible mechanisms of W. indica extract in the paper.

Author Response

Why did the authors use only Female mice?

- We appreciate reviewer’s valuable comment. Female mice are more susceptible to the development of allergic inflammation diseases than male mice [Melgert, B. N., Postma, D. S., Kuipers, I., Geerlings, M., Luinge, M. A., Van der Strate, B. W. A., ... & Hylkema, M. N. (2005). Female mice are more susceptible to the development of allergic airway inflammation than male mice. Clinical & Experimental Allergy, 35(11), 1496-1503]. Therefore, we chose female mice to evaluate anti-allergic, anti-inflammatory, and anti-AD properties of W. indica extract in this study.

The author should discuss the active components and possible mechanisms of W. indica extract in the paper.

- In our revised version of manuscript, we performed the HPLC/MS study for phytochemical analysis of W. indica extract and the result is newly added (Please see 3.6. The phytochemical characterization of W. indica using the high-performance liquid chromatography/mass spectrometry (HPLC/MS) in Results and Figure 6). According to the result of chemical profiling, two coumarins (umbelliferone and daphnoretin) and a flavonoid (quercitrin) are major components of W. indica, and those compounds might improve the skin eruption or reduce IL-4 expression. Further studies are needed to reveal possible mechanisms of W. indica extract and isolated compounds and we will do it in the near future.

Round 2

Reviewer 1 Report

The authors well addressed to my criticism.